# Machine Learning-Based Prediction of Mental Well-Being Using Health Behavior Data from University Students

**DOI:** 10.3390/bioengineering10050575

**Published:** 2023-05-10

**Authors:** Hanif Abdul Rahman, Madeline Kwicklis, Mohammad Ottom, Areekul Amornsriwatanakul, Khadizah H. Abdul-Mumin, Michael Rosenberg, Ivo D. Dinov

**Affiliations:** 1Statistics Online Computational Resource (SOCR), University of Michigan, Ann Arbor, MI 48109, USA; ottom.ma@yu.edu.jo (M.O.); statistics@umich.edu (I.D.D.); 2PAPRSB Institute of Health Sciences, Universiti Brunei Darussalam, Gadong BE1410, Brunei; khadizah.mumin@ubd.edu.bn; 3School of Public Health, University of Michigan, Ann Arbor, MI 48109, USA; mkwick@umich.edu; 4Information Systems, Yarmouk University, Irbid 72501, Jordan; 5College of Sports Science and Technology, Mahidol University, Nakhon Pathom 73170, Thailand; areekul.pua@mahidol.ac.th (A.A.); michael.rosenberg@uwa.edu.au (M.R.); 6School of Human Sciences, University of Western Australia, Perth 6009, Australia; 7School of Nursing and Midwifery, La Trobe University, Bundoora 3086, Australia

**Keywords:** mental well-being, machine learning, algorithms, university students, Asian population, health behaviors

## Abstract

Background: Since the onset of the COVID-19 pandemic in early 2020, the importance of timely and effective assessment of mental well-being has increased dramatically. Machine learning (ML) algorithms and artificial intelligence (AI) techniques can be harnessed for early detection, prognostication and prediction of negative psychological well-being states. Methods: We used data from a large, multi-site cross-sectional survey consisting of 17 universities in Southeast Asia. This research work models mental well-being and reports on the performance of various machine learning algorithms, including generalized linear models, k-nearest neighbor, naïve Bayes, neural networks, random forest, recursive partitioning, bagging, and boosting. Results: Random Forest and adaptive boosting algorithms achieved the highest accuracy for identifying negative mental well-being traits. The top five most salient features associated with predicting poor mental well-being include the number of sports activities per week, body mass index, grade point average (GPA), sedentary hours, and age. Conclusions: Based on the reported results, several specific recommendations and suggested future work are discussed. These findings may be useful to provide cost-effective support and modernize mental well-being assessment and monitoring at the individual and university level.

## 1. Introduction

Over four decades of research has linked positive mental well-being to improvements in health, development, and longevity [1]. Mental well-being can be seen as a separate, independent state from mental illness. A 10-year longitudinal study showed that improving mental well-being reduced the risk of developing mental illness by up to 8.2 times in people without mental health disorders [2]. Long term poor psychological well-being is an important indicator in developing mental illness, viz., depression, anxiety disorders, eating disorders, and addictive behaviors [3]. Thus, elevating mental well-being has become an essential therapeutic route to disease prevention.

Negative mental well-being typically manifests in young adults. However, some evidence showed that help-seeking behaviors start late when symptoms of mental illness have already appeared [4]. Help-seeking is further delayed with societal stigma, particularly in Asian cultures that often results in under-reporting cases related to mental illness [5]. Therefore, it is important to have a predictive mechanism to identify young people with negative mental well-being early to minimize the risk of developing mental health disorders [3,5].

The literature promisingly evidences that good physical health is a crucial factor influencing mental well-being [6,7,8,9]. Meta-analyses aggregating the results from numerous studies have revealed important links between mental disorders and physical inactivity and association with non-communicable diseases (NCDs) such as diabetes, heart disease, and multi-morbidity disorders [10]. Although NCDs are usually asymptomatic in young adults, it is an added benefit to promote healthy and active behaviors early to prevent or delay the development of NCDs—as these health habits track into middle age and old age. The health behaviors of young adults, particularly university students, provide important insight into NCD levels in the future [11].

As data science techniques are no longer restricted to its predecessors of applied mathematics, statistics, and computer science, it is timely and practical for social and health sciences to utilize machine learning algorithms to address issues that have profound effects on human lives [12]. In this regard, machine learning classifiers can be used to close this gap and provide more effective early detection and assessments of mental well-being in health prevention programs. Recent evidence has suggested that machine learning algorithms such as decision trees, support vector machines, and neural networks provide the most accurate prediction model for psychological issues including stress, depression, and anxiety among university students [13]. Additionally, the performance of a machine learning model and important feature selection varies significantly on data from different countries [14]. Therefore, mental well-being is directly related to the social and cultural aspects of the population in different regions [3]. Hence, it is essential to use regional data for a prediction system that is customized for the target region. This system is particularly important for the Association of Southeast Asian Nations (ASEAN) University Network-Health Promotion Network (AUN-HPN) as mental illness prevention is one of the key and immediate priorities of the network. In addition, studies on mental well-being have become a priority in higher education institutions during the COVID-19 pandemic and will continue to be important in the post-pandemic era due to the heightened risk for developing serious mental health issues [1,15]. Furthermore, the Southeast and East Asian region, where the ASEAN is inclusive, is the fastest growing digital market in the world with values exceeding US$100 billion in 2019 and is expected to grow by four times that of the regional gross domestic product by 2023 [16]. To incentivize the growing digital economy, higher education institutions should prioritize and ensure data infrastructure readiness and connectivity in the region for easement of research and development in the era of the digital revolution. Therefore, this study aims to classify negative mental well-being based on indicators of healthy behaviors among ASEAN university students using machine learning prediction models.

## 2. Materials and Methods

### 2.1. Data

The data used in this study was extracted from an online cross-sectional survey of 15,366 university students from the ASEAN countries. The target universities consisted of 17 ASEAN University Network (AUN) member universities across seven ASEAN countries, namely, Brunei Darussalam, Indonesia, Malaysia, Philippines, Singapore, Thailand, and Vietnam. The survey collects data on mental well-being and important health-risk behaviors, particularly risk factors related to non-communicable disease including physical activity and sedentary lifestyle, poor diet, tobacco use, and alcohol consumption. Demographic factors consisted of age, gender, grade point average, year of study, and living arrangement. The full description of the sample and methods can be found here [17].

Ethical approval was obtained from the institutional review board of each university prior to conducting the study (see Declarations).

### 2.2. Data Preprocessing

Data cleaning procedures were employed, including the removal of ineligible cases, duplicate responses, responses with more than 50% missing values (listwise deletion), and invalid questionnaire responses. A total of 15,366 remaining cases were used in the subsequent analysis. Missing data in these valid cases were handled using multiple imputation techniques—MICE (multivariate imputation via chained equations) using 10 multiple imputations to replace missing with predicted values using R package mice [18]. The dataset was unbalanced with respect to the binary outcome of negative or poor mental well-being. To avoid potential bias in the AI/ML modeling, the dataset was re-balanced using the Synthetic Minority Oversampling Technique (SMOTE) [19].

### 2.3. Feature Selection

According to the principle of parsimony, simplicity or a simple a priori model often provides the best explanation of a problem relative to more complex models because the inclusion of unnecessary features creates intrinsic and extrinsic noise [20]. Accounting only for key data elements avoids model overfitting, provides better predictive accuracy and generalization, and facilitates practical application [21]. Due to limitations of different types of feature selection methods, three strategies were used to validate the selection of salient variables or features that will be used in the training models in this study. The first strategy was based on the Benjamini–Hochberg false discovery rate method that controls for the expected proportion of false rejection of features in multiple significance testing [22], which could be expressed as follows:(1)$$\underset{\mathrm{False}\;\mathrm{Discovery}\;\mathrm{Rate}}{\underbrace{\;\mathrm{FDR}\;}}=\underset{\mathrm{expectation}}{\underbrace{\;E\;}}\left(\underset{\mathrm{False}\;\mathrm{Discovery}\;\mathrm{Proportion}}{\underbrace{\;\frac{\#\mathrm{FalsePositives}}{\mathrm{total}\;\mathrm{number}\;\mathrm{of}\;\mathrm{selected}\;\mathrm{features}}\;}}\right)$$

Second, a deterministic wrapper method based on stepwise selection, an iterative process of adding important features to a null set of features and removing the worst-performing features from the list of complete features, was computed [20]. The final strategy utilized a randomized wrapper method, Boruta, which iteratively removes features that are relatively less statistically significant compared to random probes [23]. Our aggregate feature selection technique utilized the intersection of these three variable elimination strategies and generated a smaller collection of variables used in the subsequent AI modeling.

### 2.4. Training Machine Learning Classifiers

Classification is a supervised machine learning technique that group records into sets of homologous observations associated with particular classes. Different classifiers or classification algorithms are available. In this study, six different classifiers were trained, including a generalized linear model (glm), k-nearest neighbor (knn), naïve Bayes (nb), neural network (nnet), random forest (rf), and recursive partitioning (RPART).

The generalized linear model, specifically, logistic regression, is a linear probabilistic classifier. It takes in the probability values for binary classification, in this case, positive (0) and negative (0) mental well-being, and estimates class probabilities directly using the logit transform function [24].

Naïve Bayes predicts class membership probabilities based on the Bayes theorem and naive assumption that all features are equally important and independent [25]. Bayes conditional probability could be expressed as follows:(2)$$\mathrm{Posterior}\;\mathrm{Probability}=\frac{\mathrm{Likelihood}\times \mathrm{Prior}\;\mathrm{Probability}}{\mathrm{Marginal}\;\mathrm{Likelihood}}.$$

Essentially, if the probability of class level *L* given an observation, represented as a set of independent features $F_{1},F_{2},\ldots ,F_{n}$, then the posterior probability that the observation is in class *L* is equal to
(3)$$P(C_{L}|F_{1},\ldots ,F_{n})=\frac{P\left(C_{L}\right)\prod_{i=1}^{n}P(F_{i}|C_{L})}{\prod_{i=1}^{n}P\left(F_{i}\right)},$$
where the denominator, $\overset{n}{\underset{i=1}{\prod }}P\left(F_{i}\right)$, is a scaling factor that represents the marginal probability of observing all features jointly.

For a given case $X=\left(F_{1},F_{2},\ldots ,F_{n}\right)$, i.e., a given vector of features, the naive Bayes classifier assigns the most likely class $\hat{C}$ by calculating $\frac{P\left(C_{L}\right)\prod_{i=1}^{n}P(F_{i}|C_{L})}{\prod_{i=1}^{n}P\left(F_{i}\right)}$ for all class labels *L*, and then assigning the class $\hat{C}$ corresponding to the maximum posterior probability. Analytically, $\hat{C}$ is defined by
(4)$$\hat{C}=\arg\underset{L}{\max}\frac{P\left(C_{L}\right)\prod_{i=1}^{n}P\left(F_{i}|C_{L}\right)}{\prod_{i=1}^{n}P\left(F_{i}\right)}.$$

As the denominator is static for L, the posterior probability above is maximized when the numerator is maximized, i.e., $\hat{C}={\mathrm{argmax}}_{L}P\left(C_{L}\right)\overset{n}{\underset{i=1}{\prod }}P(F_{i}|C_{L}).$

Artificial neural networks, or simply neural nets, simulate the underlying intelligence of the human brain by using a synthetic network of interconnected neurons (nodes) to train the model. The features are weighted by importance and the sum is passed according to an activation function, and an output (*y*) is generated at the end of the process [25]. A typical output could be expressed as follows:(5)$$y\left(x\right)=f\left(\overset{n}{\underset{i=1}{\sum }}w_{i}x_{i}+w_{o}b\right).$$

The random forest classifier is a randomized ensemble of decision trees that recursively partition the dataset into roughly homogeneous or close to homogeneous terminal nodes. It may contain hundreds to thousands of trees that are grown by bootstrapping samples of the original data. The final decision is obtained when the tree branching process terminates and provides the expected forecasting results given the series of events in the tree [25,26].

Recursive partitioning (RPART) is another decision tree classification technique that works well with variables with definite ordering and unequal distances. The tree is built similarly as random forest with a resultant complex model. However, the RPART procedure also trims back the full tree into nested terminals based on cross-validation. The final model of the sub-tree provides the decision with the ‘best’ or lowest estimated cross-validation error [27].

The caret package was used for automated parameter tuning with the repeatedcv method set at 15-fold cross-validation re-sampling that was repeated with 10 iterations [28].

In this study, random forest outperformed other machine learners. However, general decision trees might overfit model to noise in the training dataset. To overcome this, we implemented bootstrap aggregation (bagging) and boosting to reduce variance and bias, respectively.

Bagging decreases the variance in the prediction model by essentially generating additional data for training the original dataset using bootstrapping methods. Boosting reduces bias in parameter estimation by sub-setting the original data to produce a series of models and boost their performance (in this case, measured by accuracy) by combining them together [25].

### 2.5. Model Performance Metrics

Classification model performance could not be evaluated with a single metric; therefore, a number of metrics were used to assess model performance including accuracy, error rate, kappa, sensitivity, specificity, area under the receiver operating characteristics curve (AUC), and Gini index.

In binary classification, accuracy is calculated using the 2 × 2 confusion matrix, which can be expressed as follows:(6)$$\mathrm{Accuracy}=\frac{\mathrm{TP}+\mathrm{TN}}{\mathrm{TP}+\mathrm{TN}+\mathrm{FP}+\mathrm{FN}}=\frac{\mathrm{TP}+\mathrm{TN}}{\mathrm{Total}\;\mathrm{number}\;\mathrm{of}\;\mathrm{observations}}$$
where true positive (*TP*) is the number of observations that correctly classified as “yes” or “success”. True negative (*TN*) is the number of observations that correctly classified as “no” or “failure”. False positive (*FP*) is the number of observations that incorrectly classified as “yes” or “success”. False negative (*FN*) is the number of observations that incorrectly classified as “no” or “failure” [25].

The error rate is the proportion of misclassified observations calculated using
(7)$$\mathrm{Error Rate}=\frac{\mathrm{FP}+\mathrm{FN}}{\mathrm{TP}+\mathrm{TN}+\mathrm{FP}+\mathrm{FN}}=\frac{\mathrm{FP}+\mathrm{FN}}{\mathrm{Total}\;\mathrm{number}\;\mathrm{of}\;\mathrm{observations}}=1-\mathrm{accuracy}$$

The accuracy and error rate and accuracy add up to 1. Therefore, a 95% accuracy means a 5% error rate [25].

Kappa statistic measures the possibility of a correct prediction by chance alone and evaluates the agreement between the expected truth and the machine learning prediction. When kappa = 1, there is a perfect agreement between a computed prediction and an expected prediction (typically random, by-chance, and prediction). Kappa statistics can be expressed as follows [25]:(8)$$\mathrm{kappa}=\frac{P\left(a\right)-P\left(e\right)}{1-P\left(e\right)}$$
where *P*(*a*) and *P*(*e*) simply denote the probability of actual and expected agreement between the classifier and the true values.

A common interpretation of the kappa statistics includes the following:Poor agreement: less than 0.20;Fair agreement: 0.20–0.40;Moderate agreement: 0.40–0.60;Good agreement: 0.60–0.80;Very good agreement: 0.80–1.

Sensitivity is a statistic that indicates the true positive rate and measures the proportion of “success” observations that are correctly classified. This can be expressed as follows:(9)$$\mathrm{sensitivity}=\frac{\mathrm{TP}}{\mathrm{TP}+\mathrm{FN}}.$$

On the other hand, specificity is a statistic that indicates the true negative rate measures the proportion of “failure” observations that are correctly classified. This can be expressed as follows:(10)$$\mathrm{specificity}=\frac{\mathrm{TN}}{\mathrm{TN}+\mathrm{FP}}.$$

The receiver operating characteristic (ROC) curve plots the trade-off between the classification of true positive (sensitivity) and avoiding false positives (specificity). The area under this curve serves as a proxy of classifier performance and is normally interpreted as follows:Outstanding: 0.9–1.0;Excellent/good: 0.8–0.9;Acceptable/fair: 0.7–0.8;Poor: 0.6–0.7;No discrimination: 0.5–0.6.

The Gini Index (*GI*) is based on the variable importance measure and evaluates information gain by calculating the estimated class probabilities. This can be expressed as follows:(11)$$GI=\underset{k}{\sum }p_{k}\left(1-p_{k}\right)=1-\underset{k}{\sum }p_{k}^{2}$$
where *k* indexes all classes.

## 3. Results

The cleaned and preprocessed dataset comprises *n* = 15,366 cases with *k* = 20 features. The majority of respondents were from Vietnam (33.3%), followed by Indonesia (28.8%) and Thailand (25.6%). Approximately half of the respondents were female (52.6%), were 19–21 years old (66.3%), and had normal BMI (61.5%). Over half of the respondents achieved a moderate GPA of 3.3–3.9 out of 5 (69.2%) and lived off-campus (65.2%). The highest prevalence of health–risk behaviors was consumption of sugar-sweetened beverages (82.0%), followed by snacks/fast food (65.2%), low consumption of fruits and vegetables (47.8%), and high salt intake (46.0%). Insufficient physical activity levels (less than 600 MET-min/week) were observed among 39.7% of respondents. A negative or poor mental well-being level was observed among 16.7% of respondents, whereas 13.4% drank alcohol, and 8.9% smoked.

### 3.1. Feature Importance

Figure 1 illustrated ten features that are salient to the prediction model of mental well-being. This is corroborated by the error plot, variable importance plot (accuracy), and the Gini index (Figure 2). The ten salient indicators for mental well-being rank in order by importance based on health behaviors comprised of body mass index, number of sports activities per week, grade point average (GPA), sedentary hours, age, gender, salt intake, fruit and vegetable consumption, hours of sleep, and achieved recommended physical activity levels.

### 3.2. Model Evaluation

The dataset was randomly partitioned into the training set (80%) and the testing set (20%). The training dataset was used to build the classifier models using different classification algorithms including a generalized linear model, k-nearest neighbors, naïve Bayes, neural net, random forest, and recursive partitioning. The performance of the trained classifiers was then evaluated using accuracy and kappa statistics. Figure 3 illustrates the results of the trained model performance. The overall performance effectiveness of a classifier indicated using accuracy and kappa statistics showed that random forest (accuracy = 0.921, kappa = 0.788) was the best classifier, followed by k-nearest neighbor (accuracy = 0.775, kappa = 0.554) and naïve Bayes (accuracy = 0.723, kappa = 0.433).

The trained model classifiers were then applied to the testing data set to evaluate how well they predict poor mental well-being. Table 1 shows the model evaluation on testing data. With tuning parameters using the repeatedcv method set at 10-folded cross-validation re-sampling repeated with five iterations showed that random forest clearly outperforms other classifiers (AUC = 0.966). Model optimization using bagging (AUC = 0.677) did not improve the selected random forest classifier. However, boosting (AUC = 0.959) also performed similarly. Adding complementary unstructured text information to the structured data elements did not significantly improve the performance of the random forest classifier (AUC = 0.951). Such data augmentation adds more than 20 text-derived structured data elements to the standard survey features, which runs counter to the principle of parsimony.

As the top performing classifier (random forest) represents an implicit (black box) model, Figure 4 illustrates an example of a single decision tree from the aggregate forest model that illustrates one explicit classification strategy for predicting poor mental well-being as different combinations and arrangements of these features are possible in the forest. In this example, when the feature “sports” is first considered, those who were underweight, had a higher sedentary lifestyle, and had a higher-grade point average were more likely to be classified as “poor” mental well-being.

## 4. Discussion

### 4.1. Main Findings

Mental well-being is an important indicator for mental health, and this study developed prediction models using machine learning classifiers to predict the mental well-being status among university students from ASEAN countries. This is particularly important because with no end in sight of the pandemic due to the emergent of different COVID-19 variants, social and physical distancing and isolation will be the way of life in the new normal that indubitably increase the risk of developing serious mental illness in the future.

In the present study, the prediction models that produced high accuracy were achieved by random forest, random forest with text predictors, and adaptive boosting. Models using the additional text-derived features did not improve the model performance. This could be explained by the non-specific nature of the open-ended survey question regarding physical health. Future studies could examine this aspect closely using more sophisticated methods of natural language processing (NLP), deep learning, and language syntax techniques to transform the unstructured text into quantitative data elements [25]. Such advanced machine learning strategies could enhance the contribution of the textual content in the forecasting of mental well-being. Nevertheless, studies on mental well-being using various classification techniques among samples in the Asian population are scarce. A few studies also reported that random forest or decision tree-based algorithms were some of the best techniques for forecasting mental health [3,29,30]. A recent systematic review and meta-analysis has revealed that psychological interventions’ efficacy was generally low to moderate and largely concentrated in clinical mental health settings (1). The disparity of evidence for the general population can be reduced using better prediction models. The prediction models used in this study may aid in health promotion strategies and research designed to improve mental well-being, particularly from the perspective of precision health and digital health solutions.

In addition, this project also identified salient features for assessing mental well-being, namely, body mass index, number of sports activities per week, grade point average (GPA), sedentary hours, age, gender, salt intake, fruit and vegetable consumption, hours of sleep, and achieved recommended physical activity levels. Physical health and dietary status have been well-documented as strong predictors of psychological well-being [15,31]. Furthermore, interventions that target physical health outcomes have also shown to benefit mental well-being [32].

### 4.2. Limitations

The key strength of the present study is the use of a large data set from multiple sites in Southeast Asia to objectively rank the order of significant variables for predicting mental well-being. The utilization of more than one feature selection method, machine learning classifiers, and model evaluation metrics reduces the errors and biases of the results. Despite the strengths, several limitations should be noted for the current study. The data was collected using a cross-sectional survey design and is not able to draw causal inferences. Even though the survey consisted of items from widely used, validated questionnaires, self-reporting bias and likelihood of under-reporting are still present. However, this survey took place during the COVID-19 pandemic and will continue to complement and benefit future studies post-pandemic.

### 4.3. Recommendations and Future Work

The ASEAN university network (AUN) has a central role in coordinating the effective use of the available digital infrastructure and research activities, which currently focused on particular individual universities. Here are *three-point recommendations*, from general to specific, which provide feasible and practical solutions particularly for their health promotion network. *Firstly*, the continuity of data collection to train the machine learning algorithms are vital. Appropriate central data collection, processing, and analytical center using existing infrastructure, particularly from resource rich AUN member universities, need to be identified and established. Using agreed upon data collection, storage, and sharing policies and regulations encourages active participation and contribution of university students’ health data, even from resource deprived institutions. The systematic and long-term collection of health data is crucial for answering critical research inquiries, encouraging innovative interventions, and facilitating experimentation using high dimensional data created, used, and shared among higher education institutions.

*Secondly*, the selected data center that has been setup could be populated with data from affordable technological survey tools, which provide a practical, cost-effective, and long-term avenue for monitoring the overall health of university students and their mental well-being [33]. Traditional assessment tools take a considerable amount of time to complete due to the collection of many features and the collection period being over a long period before they provide useful insights [33,34]. The use of technological surveys such as mHealth mobile apps or integration into student evaluation tools that collect only a few salient features over a short period of time could have potential for more effective assessment and monitoring. Working in conjunction with the diagnostic assessment by university counselors/psychiatrists/care professionals, the collected data could provide precise early detection and early intervention [33,35]. However, ethical collection and storage of student information will necessarily require additional steps to ensure privacy and security as well as aligning with a code of professional practice such as maintaining patient–physician confidentiality [36].

*Finally*, an agreed upon schema of data will be collected and displayed as active indicators for monitoring and assessment of the current state of university students’ health and well-being status on a digital dashboard that is accessible by students and stakeholders alike. The dashboard will provide a sense of digital connectiveness among students in the ASEAN university network—the fundamental goal of the ASEAN. The data schema could be updated from time to time. This study used rank ordered importance features for predicting mental well-being, which is particularly essential in resource deprived institutions needing to allocate resources appropriately and as necessary. Among the most significant factors, the top five variables appeared to have large effects on the accuracy of the classifiers, viz., body mass index, number of sports activities per week, grade point average (GPA), sedentary hours, and age. Therefore, these factors should be prioritized when developing and implementing psychological well-being monitoring and promotion interventions. Future research is needed to continue improving the precision of the prediction models. Given that the level of mental well-being varies widely within social and cultural contexts, country or culture-specific models should be developed. In addition, resource rich institutions could consider reducing the subjectivity of the data by incorporating salient objective measures such as real-time biofeedback information from physiological sensing technology that collects electroencephalogram (EEG) activity, electrocardiogram (ECG) fluctuations, heart rate, breathing rate, temperature, speech intonation, and so on [33]. These variables are very useful in the next step of prediction model development using deep learning algorithms. Although objective measures typically will increase the cost of assessment and monitoring, university and related policymakers must decide on the trade-off between comprehensiveness and conciseness. More importantly, to make this a reality, the AUN-Health Promotion Network is in a central position to coordinate and collect long term monitoring data for all member institutions in the ASEAN region [37]. The continuous collection of data is critical for the organization to modernize and adopt data analytics to improve operational efficiency and achieve strategic goals.

## Figures and Tables

**Figure 1 bioengineering-10-00575-f001:**
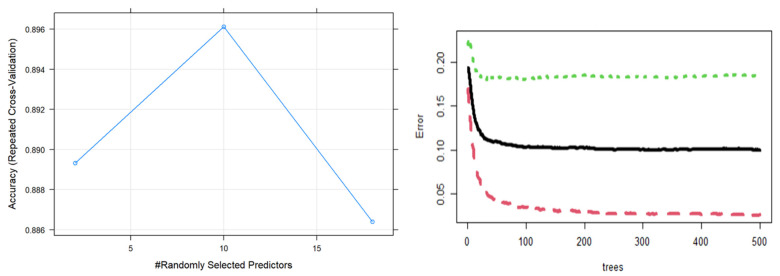
Accuracy (**left**) and error (**right**) plots of random forest classification of mental well-being. In the error-trees plot, the green and red dotted lines indicate the upper and lower bounds of optimal trees for the model, respectively.

**Figure 2 bioengineering-10-00575-f002:**
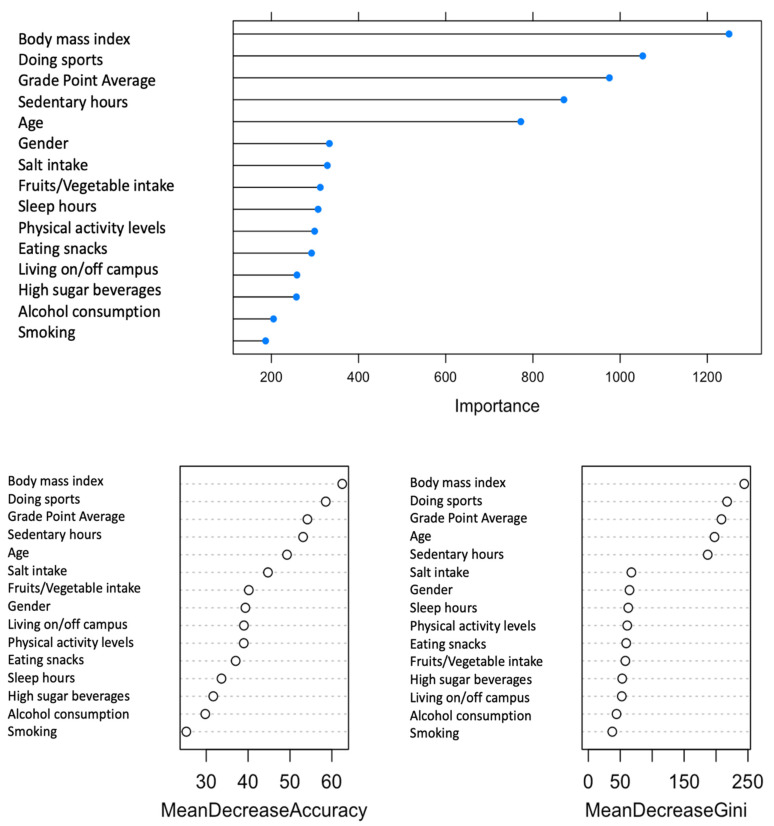
Variable importance plots of random forest classification of salient features (**top**) of mental well-being using accuracy (**left**) and the Gini index (**right**) as evaluation metrics.

**Figure 3 bioengineering-10-00575-f003:**
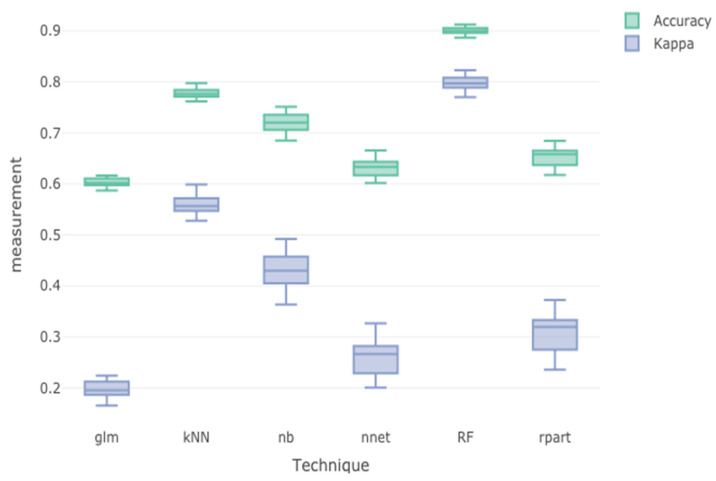
Evaluation of performance of trained machine learning classifiers.

**Figure 4 bioengineering-10-00575-f004:**
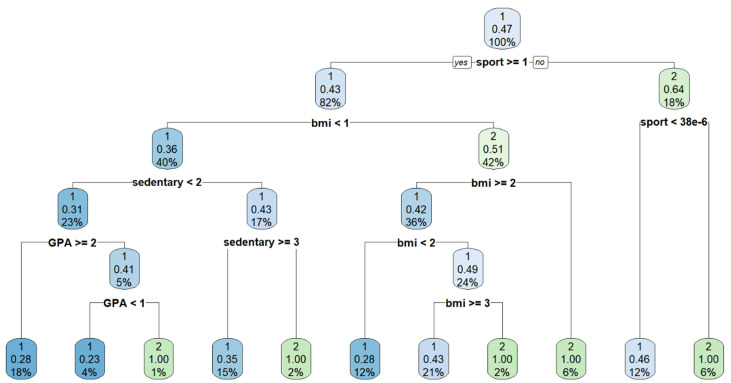
An example of a decision tree with a recursive partitioning algorithm for predicting negative mental well-being. The blue colored nodes represent a higher correlation or chance to be classified as poor mental well-being. In contrast, the green nodes represent lower probability of being classified as poor mental well-being. The percentage in each refers to the proportion of cases remaining after each split—starting with 100% from the top of the hierarchy.

**Table 1 bioengineering-10-00575-t001:** Model evaluation metrics of machine learning classifiers on the testing data set.

Classifier	Accuracy	Kappa	Sensitivity	Specificity	AUC
Random Forest	0.901	0.801	0.980	0.815	0.966
Random forest + text	0.881	0.759	0.965	0.782	0.951
Adaptive boosting	0.893	0.785	0.951	0.828	0.959
k-nearest neighbor	0.795	0.593	0.711	0.887	0.886
Naïve Bayes	0.702	0.399	0.775	0.621	0.678
Bagging (Bootstrap Aggregation)	0.672	0.349	0.617	0.735	0.677
Recursive partitioning	0.633	0.268	0.607	0.662	0.665
Neural Network	0.615	0.231	0.597	0.633	0.674
Generalized linear model	0.603	0.201	0.673	0.527	0.653

## Data Availability

The datasets generated and/or analyzed during the current study are not publicly available due to restrictions on intellectual property regulations of the funding organization.

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
