# Peer review of "Machine Learning-Based Prediction of Mental Well-Being Using Health Behavior Data from University Students"

_bioengineering, 2023, doi:10.3390/bioengineering10050575_

Round 1

Reviewer 1 Report

This paper uses machine leanring techniques to classify negative mental well-being based on indicators of healthy behaviors among university students, which includes data pre-processing, feature selection, and classification. This work is generally a good reference for similar work among broad areas. I have several comments as follows:

1) Works on applying machin learning in the classification of mental well-being should be reviewed in the introduction section.
2) Detailed information on the considered variables related to mental well-being should be given in Section 2.1, in addition to a reference, such as the total number and a full list of all factors.
3) Some typos in line 184 and 214.
4) I think readers would be interested in how to reveal useful knowledge according to the analysis results. Therefore, I suggest to give detailed explanation on typical results. For example, more explanation should be given for Figure 4. Why is the root node 'sport' since it is the second important factor according to the analysis results in feature selection step. What interesting information can be found in this hierarchical classification result.

The English language needs moderate editing as some errors occur.

Author Response

Thank you for your kind review and valuable comments. We highly appreciate it. Please kindly find the point-by-point response to your comments below. Corresponding changes have been made in the attached manuscript, in red font.   1) Thank you for the suggestion. Important details of the of the application of machine learning on mental wellbeing in recent articles, is added to the introduction.   2) Variables and total number of cases in the data, is added to the Data section.   3) Typo errors corrected.   4) The reviewer is correct – more details are added for Figure 4.

Reviewer 2 Report

1.      various machine learning algorithms such as generalized linear models, k-nearest neighbor, naïve-Bayes, neural networks, random forest, recursive partitioning, bagging, and boosting. They conclude that Random Forest and adaptive boosting algorithms achieved the highest accuracy of identifying negative mental well-being traits.

2.      The article is very well structured and very well illustrated. The results seem to be very convincing.

3.      However, the number of tests and dataset used is very limited. Better to add other datasets and test them against other models in the literature.

4.      In addition, it is preferable to propose a common histogram which groups together all the methods which have been used with different colors.

5.      Propose other metrics such as F1-measurement, UOC, etc. in order to better compare the models.

6.      Update your bibliography.

7.      Redo the summary and state of the art part which is insufficient. Other models are to be mentioned such as ensemble models, deep learning, and other models.

8.      Quote the evaluation criteria and the choices of the different attributes before comparing.

Author Response

Thank you for your kind review and valuable comments. We highly appreciate it. Please kindly find the point-by-point response to your comments below. Corresponding changes have been made in the attached manuscript, in red font.   1) Thank you. 2) Thank you. 3) Thank you for the suggestion. The researchers agree that external validation of the ML model on different datasets is ideal. However, access to relevant data is limited and is a common issue faced by any researchers. 4) Thank you for the suggestion. Considering that histogram is mainly used to visualize numerical scores or values, it would not be relevant in this case. 5) We agree, there are many metrics available and we need to include important ones – we have included Gini index, Accuracy, Kappa, Sensitivity, Specificity, and AUC. 6) Bibliography and reference updated. 7) This comment is unclear – we do not use deep learning in this study. Appreciate your clarification. 8) We are trying to strike a sensible balance between the need for succinct presentation, sufficient technical details, and informative descriptions that facilitate result reproducibility. If the reviewers recommend, we can move some of these technical details in an appendix section. Thank you.